# A Practical Perspective for a Conservative Estimate of Blood Glucose Level during Restaurant Dining and Supermarket Shopping

**DOI:** 10.3390/foods10020444

**Published:** 2021-02-18

**Authors:** Xiao Dong Chen, Peng Wu

**Affiliations:** Life Quality Engineering Interest Group, School of Chemical and Environmental Engineering, Soochow University, Suzhou Industrial Park, Suzhou 215123, China; xdchen@mail.suda.edu.cn

**Keywords:** conservative estimate, blood glucose level, *in vitro*-*in silico* prediction, food labeling, Glycemic Index, practicality

## Abstract

Foods today are so diverse and enjoyable, making healthy choices difficult. In this perspective, an *in vitro*-*in silico* approach for obtaining a conservative estimate of the postprandial blood glucose concentration, which is a realistic estimate nevertheless, after intake of a certain portion of meals is proposed. The rationales and feasibilities of the approach are described and discussed to an extent. The key idea is to first measure the maximum amount of glucose released in an *in vitro* test under standardized conditions from a specified serving size of a meal or dish or a packaged product sold in a supermarket. The value can then be translated by a *literate* consumer to the highest estimate of blood glucose rise prior to purchasing or eating through an established *in silico* blood glucose prediction model in the medical field. The strategy proposed here would help health conscious (diabetics included) and other life quality conscious individuals to make quantitative decisions on consuming the portions of different foods of desire. This strategy may be more effective in reality compared to the conventional GI (Glycemic Index) and GL (Glycemic Load) concepts.

## 1. Introduction

Unlike the advancement in animal nutrition for feed production and applications, the understanding of human nutrition cannot be easily transferred to practice as humans are highly individualistic. 

Enjoying great foods, their tastes, and flavors, is one of the most important aspects of life quality (LQ). Once with a diabetic condition, the LQ of the individual deteriorates, which may be caused by anxiety, but more likely due to the lack of means to control the blood glucose level on a daily basis. Wise eating with a diabetic condition in a common society does present great challenges. Besides medication, including insulin administration, it is known now that the intake of the food materials that can be ‘converted’ to blood glucose, starch, for instance, has to be closely monitored, ‘measured’ or ‘gauged’ by the individual. This knowledge is crucial because frequent high blood glucose concentrations are known to be closely related to the increased risks of many chronic diseases such as type II diabetes, cardiovascular disease, and coronary heart disease [1,2]. However, when one sits down to order a meal in a restaurant, how to ‘gauge’ the blood glucose rise from the meal becomes a difficult task. If a simple and effective ‘gauging’ is available in daily life, maintaining a good life quality will be more feasible. 

Previously, a significant concept called the Glycemic Index (GI) emerged, which was first proposed by David Jenkins in 1981 [3]. GI is used to emphasize the ‘ranking’ of individual food sources such as a watermelon, a potato, a carrot, etc., intending to assist people (particularly diabetes patients) in choosing less glucose-converting food. GI is generally defined as the percentage of a test food containing 50 g of available carbohydrates that will cause an increase in the incremental area under the blood glucose response curve (AUC) over a period of time (typically 2 h after a meal) compared to a comparable amount of glucose. GI is considered an important indicator of carbohydrate quality, which implies how fast (dramatically, moderately, or slightly) a carbohydrate will raise blood glucose levels compared to the reference food (i.e., pure glucose) [4]. A substantial number of epidemiological and interventional studies have reported the health benefits of lower-GI diets in losing weight, lowering fasting glucose and insulin levels, and reducing the risk of type II diabetes mellitus and cardiovascular disease [2,5,6,7]. 

The concept of the Glycemic Load (GL), initially introduced by researchers at Harvard University [8], is the term used to allow comparisons of the overall effect of the two factors on the blood glucose level elicited by realistic portion sizes of different foods. Thus, the GL combines both the quality and quantity of the carbohydrates to predict the glycemic response after food consumption. This can be particularly important for foods that are ‘healthy’ but have a high GI, such as carrots and watermelons. The GL is usually calculated by multiplying the mass (in grams) of carbohydrates contained in a serving and the GI of the food and dividing by 100 [8,9]. GL and GI are similar concepts, except that serving size is considered for calculation of the GL. Having said the above, one can see that it is not straightforward to employ the concept of GL. GI seems simple but is more for a ‘ranking’ rather than a practical method of going about daily life. 

For someone with diabetes, it is suggested to avoid high peaks in their blood glucose levels. One way to achieve this is to eat more low GI foods that are slowly digested and absorbed, producing only a slow or gradual rise in their blood glucose. Note that the GI of the same food material may vary significantly due to many food-related factors. These factors include differences in particle size, physical form, ripeness, cooking and food processing, other food components (for example, fat, protein, dietary fiber), and starch structure [7,10,11]. With respect to controlling sugar intake, in real life, these factors certainly complicate the practical use of scientific concepts such as GI. This makes it not readily possible to determine what portion of the same packaged product on the shelf of a supermarket should be consumed each time by ordinary consumers and even health professionals [4,12]. 

Conventional measurements of glycemic response curves and blood glucose concentrations are performed through frequent blood sampling after ingestion of carbohydrate products or meals. However, this method is time-consuming, highly variable between intra- and inter individuals, and associated with ethical restrictions [13]. Obviously, this method is not applicable for individuals to make quantitative decisions on consuming the portions of different foods of desire during restaurant dining and supermarket shopping daily life. After years of research in the related areas, as well as observing the practical aspects, we have come to this recommendation. This recommendation is perhaps a more realistic scheme of helping consumers to estimate the ‘risk’, which is more directly related to blood glucose rise after eating. The scheme may be viewed as practical labelling, which is so important in food-related health management issues by providing a credible source of information to consumers [5,14]. To this end, an approach for a conservative estimate of the glycemic response is proposed here. The aim of this approach is to be able to ‘explicitly’ and ‘easily’ help people (diabetics or not) to make healthier decisions regarding the portions to eat each time when ordering and purchasing meals or food. 

## 2. Rationales and Implementation of the New Approach

### 2.1. Overview of the New Approach

As stated earlier, for better control of the blood glucose level, it is very important that we can translate how much of the food (in grams or in proportion labelled in each dish on the restaurant menu or in each package on the packaged food product) that we consume would end up increasing the blood glucose and the level of this increase. This appears to be a single best information for a *literate* consumer (diabetic person included) to know to make his or her decision on how much they ought to eat. The key idea proposed here is first to determine the maximum amount of glucose released from a specified serving size of the meal or the dish or the packaged product under standardized *in vitro* digestion conditions. The value can then be used as input data for an established mathematical (*in silico*) blood glucose prediction model to give the expected yet conservative estimate of the blood glucose level (a realistic estimate nevertheless) at the time of purchasing or eating. Figure 1 is created, providing the relationships between the measured *in vitro* convertible glucose and the relative glucose area (RGA, a) or the peak glucose concentration (PGC, b) after intake of a specified serving size of a meal or food product in healthy and diabetic subjects. Figure 1a,b can be made into a chart for consumers to recognize its usefulness. It may also be useful to print on the product package or restaurant menu in actual values. 

To illustrate the rationale of this perspective, an example is shown in Figure 2 for a small yet delightful local restaurant in the ancient city Quanzhou in Fujian Province in China. The restaurant specializes in home-made starch variety-based foods. These are very tasty and traditional foods for the locals. The starches used for making different dishes are different, and the cooking methods/procedures are different as well. It is difficult to predict the GI or GL concepts for this kind of food outlet. Of course, this would also be common in other cultures. Following the suggestion in the current perspective, the product information, such as restaurant meals corresponding to Figure 2, can be made in a format that can be easily understood by a *literate* customer. The product information of the expected blood glucose rise can be shown on per serving basis (grams). This is perhaps the most reliable measure that informs a *literate* individual of what is going on after eating. 

### 2.2. Implementation of the New Approach

The new approach suggested here mainly consists of three successive steps for obtaining the conservative estimate of the blood glucose rise after the intake of a specified portion of one dish or a food product (Figure 3).

#### 2.2.1. Measuring the Maximum Amount of Glucose Released after *In Vitro* Digestion

Technically, an accurate measurement of the *in vitro* converted glucose amount from each meal on the menu or each packaged food product on supermarket shelf should not be an issue if food products are prepared through specific cooking procedures with specific ingredients. It is reasonable to claim that the maximum amount of glucose released from a specified size of a carbohydrate meal or food product under standardized *in vitro* enzymatic digestive conditions should be similar among different testing technicians provided with reasonable training. Here, we only consider the final amount of glucose released, whereas the specific digestion kinetics such as the solid disintegration and enzymatic hydrolysis rate during digestion are not considered. This is advantageous in avoiding the complexity of measuring the complex digestion kinetics to ensure consistency between intra- and inter laboratories, thereby practicing the current perspective in reality. Note certain unique local sauces are often added in some complex meals depending on the cultural origin. For instance, Chinese soy sauces are of many types depending on the fermentation conditions and the ingredients. These sauces can induce complexity in measuring *in vitro* converted glucose. For example, the color of sauces is undesirable for the commonly employed 3,5-dinitrosalicylic acid (DNS) method that relies on measuring the refractory index [17]. Upon separating the solid materials (insoluble in the digestive fluids used *in vitro*) by centrifugation, the supernatant can be sampled and diluted (until the effect is minimal) to reduce the impact of the coloring on the refractory index measurement. However, many dilutions would reduce the accuracy of the method. As such, the high performance liquid chromatography (HPLC) method [18] that has been widely used for sugar quantitation [19,20] may be preferred, which is not theoretically affected by the co-existence of many ingredient compounds. Of course, quantitation needs to be carried out carefully to ensure that the kind of accuracy required can be achieved. 

#### 2.2.2. *In silico* Prediction of Glucose Glycemic Response Curves

The *in silico* model is introduced to enable the translation of measured values of the glucose released from a given dish or a packaged ready-to-eat product, obtained from the *in vitro* digestion tests as measured in the above section. Then, the plasma glucose concentration curves (glycemic responses) for healthy and diabetic subjects can be obtained. A few mathematic models of the glucose–insulin feedback system have been proposed in the literature for various purposes, i.e., to understand the mechanism of glucose homeostasis, to analyze experimental data, to identify and quantify relevant biophysical parameters, to design clinical trials, and to evaluate diabetes prevention or disease modification therapies [21,22]. The *in silico* model used here is a spatiotemporally distributed intestinal absorption model coupled to an established glucose–insulin regulation system. This *in silico* blood glucose prediction model has been validated against human clinical data with regard to the accurate prediction of human glycemic response curves following the absorption of glucose in healthy and diabetic subjects [15,22,23]. The *in silico* model integrates the main gastrointestinal processes including gastric emptying, transport along the small intestine, and absorption through the intestinal membrane. The origin and mechanism of the *in silico* model have been detailed in recent work [23]. With the measured *in vitro* convertible glucose as input data for the *in silico* blood glucose prediction model, the relative glucose area (RGA), and the peak blood glucose concentration (PGC) in mmol/L based on the predicted glycemic response curves can be obtained after ingesting a specified serving size of a dish or a food product. Thus, a working chart as shown in Figure 1 can be translated in explicit formats printed on restaurant menus or supermarket product packages. 

#### 2.2.3. Food Labelling to Practice the New Approach

Restaurants may be advised to take the approach, which can be done to provide their customers with much better information. Following the suggestion proposed in this perspective, for a given dish or a packaged ready-to-eat product (listed in *Y* grams), the *in vitro* convertible glucose in *X* grams per dish or per product can be labelled on the menu or package. Based on the established working chart, one can easily work out how much a portion of *Y* may be consumed to raise blood glucose to the level of *x* (mmol/L) as would be detected using a commercial personal blood glucose meter. The value *x* can also be directly expressed on the menu or the package, which may be easier to understand than the chart. For a dish such as ‘Sweet and Sour Pork’ which is fairly popular in China and elsewhere as a famous yet common Chinese cuisine, a health-conscious restaurant can standardize their menu as Sweet and Sour Pork (300 g, *X* grams of converted glucose, *x* (mmol/L) in blood). It says eating it all, i.e., 300 g, one would have *x* (mmol/L) peak value in the blood stream of the consumer. It is well known that in making this dish, corn flour and sucrose (sometimes honey) are involved. The cooking procedure involves dipping pork pieces (which can have various levels of fat) in corn flour slurry and then frying in cooking oil to create the crispy casing. Soy sauces and vinegar are used as well. Plenty of unknowns in the making can be quantified for the purpose of calculating the glucose rise in blood. As such, this kind of labelling or indeed standardized cooking (fixed *Y* and the proportions of each ingredients used) plus the standard *in vitro* digestion test to measure the converted glucose to *X* to yield *x* for the specific dish *Y* would be the more scientific way of informing consumers. 

One can figure out one dish’s *x* (mmol/L), i.e., *x_i_* (*i* represents dish number). For a total of *N* number of dishes chosen by the customer or his/her host for the meal, for the personal choices of the fractions $\alpha_{i}$, it is possible to estimate the total *x* by calculating the total amount of *in vitro* converted glucose after consuming various fractions of the dishes ordered:(1)$$x=\sum_{i=1}^{N}\alpha_{i}x_{i}$$

In a slightly formal dinner in China, one would expect to have at least four courses, and for a really formal dinner, it would be of 10–12 dishes. A restaurant menu, in China or elsewhere, would have a dozen or several dozens of different dishes to choose from. The current approach is thus recommended to be a future means to control the blood glucose content, whilst giving the diabetic patients wise options of eating through a more scientific yet enjoyable approach. Furthermore, with the adoption of computer applications at restaurants (such as digitalized ordering, etc.), this approach can be readily programmed into a more convenient operation. The use of digitalized ordering technology will greatly help consumers who may lack education to quickly calculate the total *x*. Despite the approximate nature outlined above, the approach suggested is simple and is expected to be much more practical.

### 2.3. Uncertainty of the Approach 

The above-mentioned approach is not complex. However, uncertainty exists, at present, in at least the following areas:

The accurate account of the mass of a dish in a restaurant is necessary. The addition of the ingredients may also need to be followed by the chiefs. A repeatable cooking procedure needs to be consistent; this aspect is, however, less of an issue with the standard packaged product. 

The food products we refer to as purchased from a market are the ready-to-eat foods without the need of further processing such as cooking. The approach proposed here is not suitable for raw or semi-manufactured food products, which need further processing before they can be consumed. 

The accurate measurements of *X* of a real dish or a packaged food product, especially those so called ready-to-eat meals, have to be guaranteed.

The perceived fraction *α_i_* by the consumer himself/herself is most likely *ad hoc*, though it is not a bad choice as a practical measure. Judgement can be improved with time if the person of concern is more serious about the matter. 

## 3. Concluding Remarks

Controlling the intake of glucose-converting food materials has long been recognized as an effective means of maintaining the blood glucose level of a diabetic patient. So far, it is still not easy to practice the scientific knowledge for individuals to follow in daily life especially in many cultures where the cuisines are so diverse, and the cooking procedures are so unique. It is suggested that *in vitro* digestion measurements should be carried out for each recipe that is suspected of having high glucose-converting materials to provide the ‘generically maximum’ convertible or extracted glucose content. Upon an approximate but nevertheless effective conversion of the *in vitro* to the *in vivo* blood glucose level (usually indicated on a blood glucose meter), a commoner can be guided to make a choice of consuming a portion of a dish to his/her delight without a potentially large health hazard. It is suggested here for any cooked dish (the food mass is labelled in grams), the *health-conscious* restaurant owners can, through the tests provided by a standard laboratory with expertise in the area, obtain the desired information (here, it is the maximum extractable glucose content) and label this information on the actual menu. The same can be done for supermarket foods including packaged and non-packaged foods. In addition, it may be interesting for the providers of foods or food products to package foods in smaller units or make smaller dishes in restaurants, which is helpful to practice the strategy provided in this paper in reality. Furthermore, some other health information such as the undesirable lipids, etc., may also be listed for the benefits of preventing or managing metabolic chronic diseases, e.g., *dyslipidemia* and fatty liver in disease-prone individuals. The above was written for combating diabetic conditions in particular, but similar practical reasoning may also be beneficial for people who are interested in weight control or in competitive sports. Finally, with the development of digital technology, it is foreseeable to have the above approach implemented in a much more convenient manner. This will achieve more than GI and GL practicality yet follows the GI and GL concepts in a scientific way.

## Figures and Tables

**Figure 1 foods-10-00444-f001:**
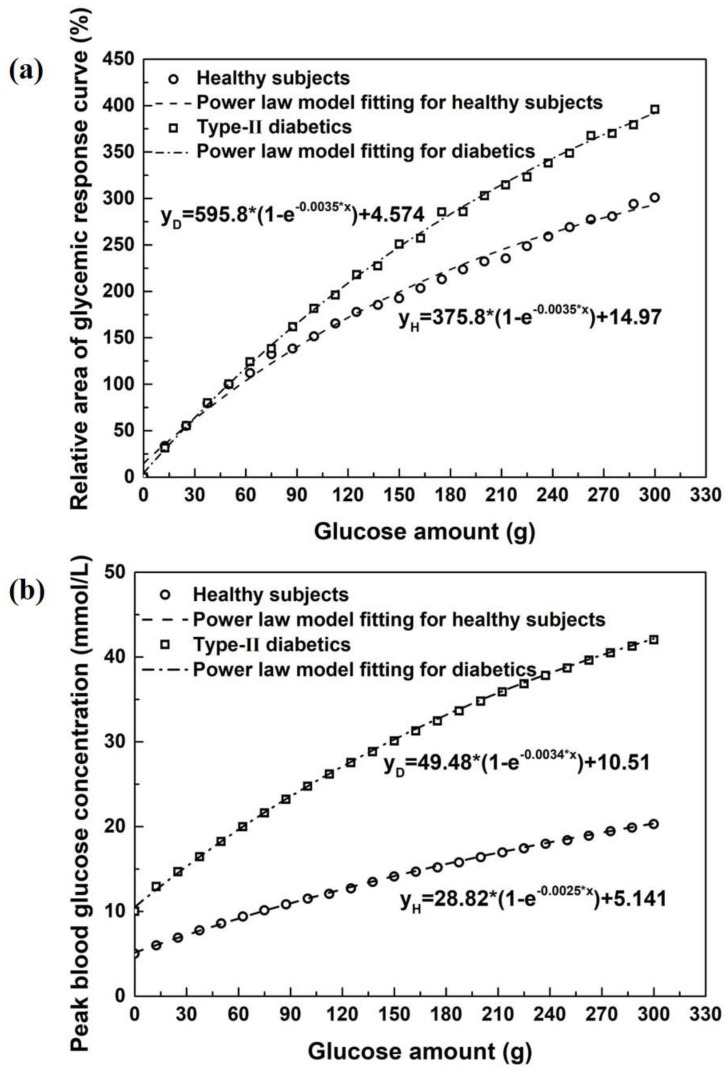
The relative glucose area (RGA, **a**) under the glycemic curve (defined as the net area under the glucose curve after ingestion of a specified serving size of a meal or a ready-to-eat food product divided by the net area under the glucose curve after ingestion of 50 g glucose) and the peak blood glucose concentration (PGC, **b**) in healthy and diabetic subjects. The peak glucose concentration in mmol/L is very useful as it directly corresponds to the reading of the value on a standard glucometer. The RGA and PGC are derived from the glycemic response curves, which are predicted based on the spatiotemporally distributed intestinal absorption model coupled with an established glucose–insulin regulation system [15,16].

**Figure 2 foods-10-00444-f002:**
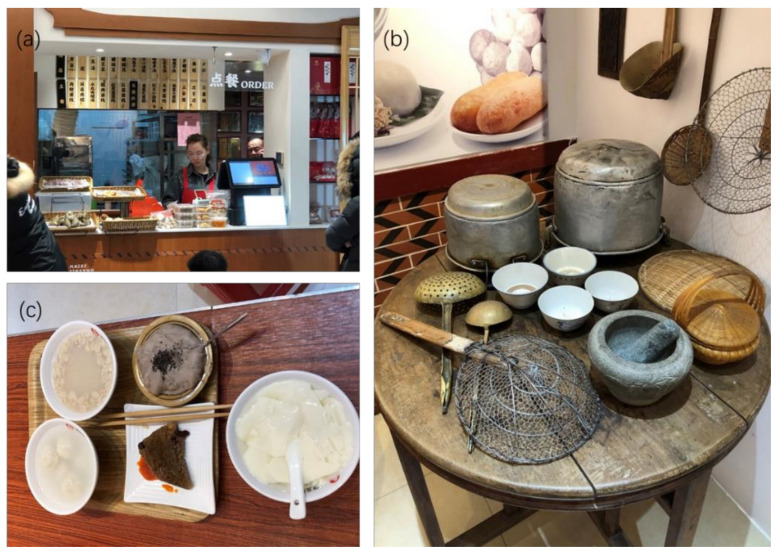
An illustrative example of ordering in a traditional restaurant that presents excellent tasty foods yet lacks information for health-conscious customers (to make better options on the sizes of the dishes and what portions of each may be consumed to make up a ‘total’ increase in the blood glucose level after the meal) (photographs taken by the 1st author). (**a**) At the counter of the particular restaurant, the customers can see the wooden plates hung at the back (upper area of the photo), which are marked with the names of the dishes, to order the foods, showing the basic starch origins, such as glutinous rice, taro, wheat, black bean, green beans, etc. (**b**) As a part of the effort of showing the history of the restaurant, some of the traditional tools of making these tasty foods such as a stony grinder, sieves and pots, etc. One can imagine that the cooking facilities and the cooking procedures and flavor ingredients would be different too. (**c**) Some of the foods on a table in the restaurant as ordered by the first author, which are the favorites of this author.

**Figure 3 foods-10-00444-f003:**
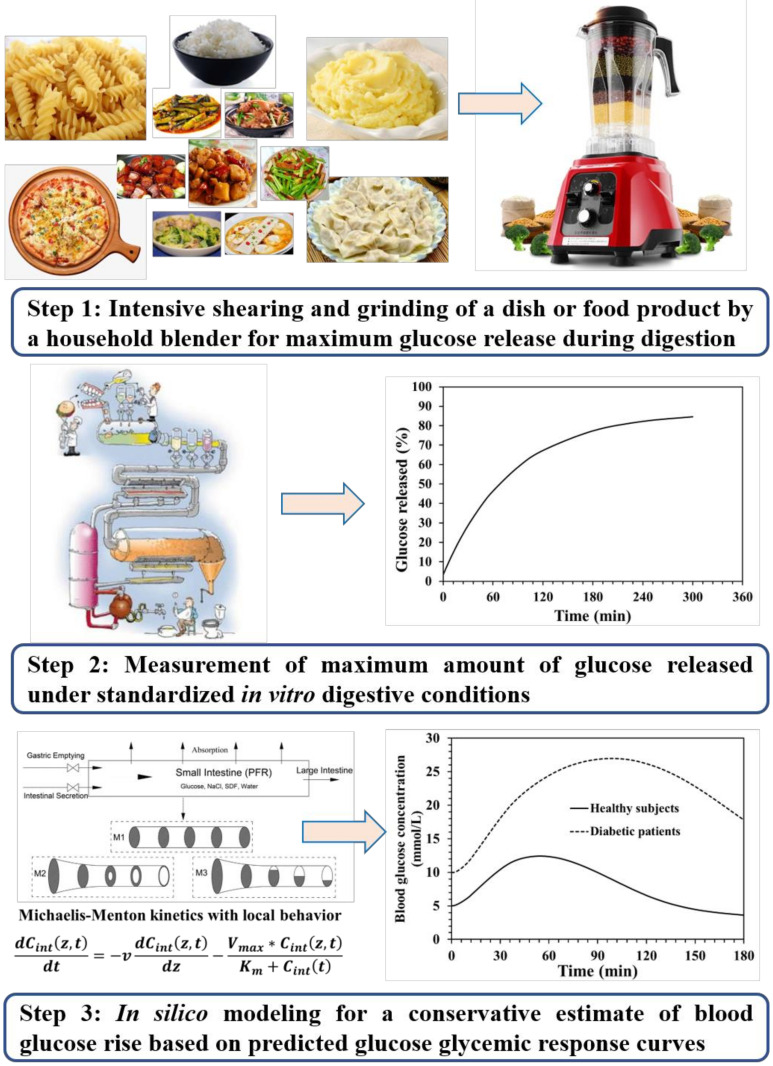
An overview of the *in vitro*-*in silico* modeling approach for a conservative estimate of glucose response curves after the intake of a specified serving size of a dish or a ready-to-eat food product. In general, this approach can be divided into three steps: (**1**) pretreatments of a specified size of dish or food product to be digested *in vitro*, which may include intensive mixing and grinding by a household blender to destroy the food structures, thus maximumly facilitating the release of glucose during digestion; (**2**) measurement of the maximum amount of convertible glucose released from the food tested under standardized *in vitro* digestive conditions; (**3**) application of the *in silico* glucose–insulin regulation model to obtain a conservative estimate of the blood glucose rise with the measured maximum *in vitro* convertible glucose as input data. Assuming this convertible amount will be processed as one glucose sample in the digestion tank; it presents a maximum possible blood glucose rise. This is viewed as a conservative indicator for a consumer.

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
