# Peer review of "A Practical Perspective for a Conservative Estimate of Blood Glucose Level during Restaurant Dining and Supermarket Shopping"

_foods, 2021, doi:10.3390/foods10020444_

Round 1

Reviewer 1 Report

Reviewers’ comments

Journal: Foods

Manuscript ID: foods-1057418

Title: “A Practical Perspective for Conservative Estimate of  Blood Glucose Level during Restaurant Dining and Supermarket Shopping"

Author(s): Xiao Dong Chen and Peng Wu

Recommendation: Accept after minor revisions.

Comments:

This manuscript has demonstrated a comprehensive study on the conservative estimation of blood glucose level during restaurant dining and supermarket shopping. The topic is very impressive as overeating of foods particularly in the urban area is very common which is closely related to high blood glucose and increased risks of many chronic diseases such as type II diabetes, cardiovascular disease, and coronary heart disease etc, but least studied. The authors have proposed the strategy which would help health-conscious (diabetics included) and other life quality conscious people to make a quantitative decision on consuming the portions of different foods of their desire. They also emphasised that the current method might be more effective compared to the conventional GI (Glycemic Index) and GL (Glycemic Load) concepts. Overall, this is a very well-prepared manuscript. 

Most of the sentences are overly long particularly in the abstract and conclusion section and author(s) should recast those long sentences into multiple short sentences for more clarity to readers. Author(s) should carefully go through the manuscript including the title of the manuscript to remove unintentional mistakes and grammatical error.

Author Response

We highly appreciate the positive comments and valuable suggestions on our manuscript. As suggested, we have divided those long sentences into several short ones throughout the manuscript. Also, the unintentional spelling and grammar errors have been corrected.

Reviewer 2 Report

The manuscript of Xiao Dong Chen and Peng Wu delay the possibility of improving the communication to the informed consumers about the nutritional values of the serviced foods in restaurants and in the markets. The argument is very intriguing, but the purposed strategy and its applications are a bit confused. At first, the food consumer and the producer must be deeply "informed" to operate under these concepts to really standardize the composition and the intake of a specific nutrient in relation to peculiar actual or potential health risks. Under this point of view, the applicability of the approach proposed in the manuscript is necessarily limited to a scarce number of individuals, and no emendation to this limit of the strategy are provided. It must also be underlined that in a food label not only a few peculiar nutritional characteristics of a product but all the bromatological specificities with particular regards for the risks derived from its assumption should be described. The ethic-producer can not represent only the aspects that can be interpreted by the consumer uniquely under a healthy-positive point of view. For similar reasons, the amount of nicotine and tar are recently removed from cigarettes packs in Europe (this could suggest that a trademark of cigarettes as healthier than other) by substituting this misleading information labels with images clearly indicating the health damages deriving from smoking. Likewise, a portion of fried "Sweet and Sour Pork" cannot be presented as healthier than another, in a specific menu context, in the reason of the total glycemic index or load.

Author Response

We thank the reviewer for his/her interesting comments. Yes indeed one should not attempt to label quantities of certain nutrient (or a harmful component like nicotine in cigarette) only to indicate this food may be better than the other in a speculative manner. Our intention is not complex. The work intends to inform people that if one decided or wanted to enjoy Sweet and Sour Pork, then a portion amount (based on prior evaluation and computation, and labelled on the dish menu) is recommended beyond which it may cause a concern in rising blood glucose. If a diabetic person likes to quantify/control his/her intake based on science, the approach shown in this paper would be a nice approach. This is not to say by labelling Sweet and Sour Pork would appear better than a Sweet Pork for instance. The label is intended to indicate the amount of consumption rather than showing one is better than another. Of course if the conversion to glucose is less than another, one can enjoy eating more of this product.

Following the arguments provided in this paper, it may also be interesting for the providers of foods or food products, to package foods in smaller units or make smaller dishes in restaurants well informed by the data on these foods or products (Lines 344-347).

Finally, the idea shown in this paper, once adopted one way or another in practice, will help many people. Over 10% of the populations in China alone are diabetic. The diabetic population in China and that in India combined, makes the world’s 3rd largest country as such.

We have gone through the paper again to make sure the above idea has been put forward with little confusion.

Reviewer 3 Report

Figure 3: A figure should be able to stand alone. Readers do not need to go back and forth between the figure and the text. Could you please update the figure? For instance, could you please explain the step 1? The link between the foods and the mixer is not clear. Are you going to mix the dumplings, pizza by the mixer?

Line 260: The foods purchased from the market (‘food as bought’) may not be the ‘foods as eaten’/ ‘food as consumed after preparation’ in terms of dry or wet basis, after frying, cooking, etc.  Comments on this here? So how to overcome this?

Line 288-293: a certain amount of diabetes patients (e.g. people lack of education) may not be able to calculate by themselves. What’s your suggestions regarding this issue? Any solutions?

Line 288-293: Could you please briefly discuss the overall feasibility of the method as well?

Line 294-296: In China, people always share dishes when eating outside. In this case, the information per dish might be useless? Any extra information should be provided?

Line 314-316: in China, chiefs rarely weigh ingredients before cooking. It could be one of the hurdles. Could you please briefly describe the fact and solutions?

Line 327: are so unique

Line 340: Same food from different areas may have very different food composition. Is it necessary to test and identify the differences? On which level the differences should be clarified?

Line 341: food labelling is always about selection of information and information priority analysis. Could you please briefly describe why the information regarding diabetes should be presented compared to other health information for other diseases?

Author Response

Thanks for the comments. As suggested, this figure has been updated with additional descriptions for the figure legend to make it more readable and easily understandable. Step 1 refers to the pretreatments on a specified size of the dish or the food product to be digested in vitro, which may include the intensive mixing and grinding by a household blender to destroy the food structures thus maximumly facilitating the release of glucose during digestion. Whether pretreatments are needed or not depends on the food species and their physicochemical properties. For solid foods such as dumplings and pizza, it is necessary to grind them into a fine homogenous system by a household blender prior to in vitro digestion test. This can maximumly destroy their food structures and ensure more thorough hydrolysis of starch to glucose during the in vitro digestion under standardized digestive conditions.

It is noted that the food products we refer to as purchased from a market are the ready to eat foods without the need of further processing such as cooking. The approach proposed here is not suitable for the raw or semi-manufactured food products, which need further processing before they can be consumed. For better clarification of this issue, we have added these sentences as a separate description of the uncertainty of the approach in 2.3 (Lines 313 to 317).

It has to say that the calculation of the rise of total peak blood glucose to the level x after consuming various fractions of the dishes ordered is actually a very simple task. For a given dish such as Sweet and Sour Pork, the information including the Y grams of dish, the X grams of maximum converted glucose, and the rise of peak blood glucose to the level x can be simply and clearly printed on menu as: Sweet and Sour Pork (Y grams, X grams of converted glucose, x (mmol/L) in blood). Based on this simple formula, for the calculation of total x, one only needs to know the fraction of each dish she/he consumes ??, which depends on personal choices. For the consumers that may lack of basic education, this calculation can be done with the aid of digitalized ordering technology. With the development of digital technology, it is foreseeable to have the proposed approach implemented in a much more convenient manner (Lines 354-356).

We have added a few more discussions about the overall feasibility of the approach as follows in lines 297 to 304: “Furthermore, with the adopting of computer applications at restaurants (like digitalized ordering etc.), this approach can be readily programmed into a more convenient operation. The use of digitalized ordering technology will greatly help the consumers who may lack of education have a quick calculation of the total x. Despite the approximate nature outlined above, the approach suggested is simple and is expected to be much more practical”.

The information printed on menu is definitely still essential and meaningful because the overall/total x of a given dish is constant although this dish may be consumed by multiple people. As already indicated in lines 321 to 324 “The perceived fraction αi by the consumer himself/herself is most likely ad hoc”, which depends on personal choice. The better judgement of the fraction of each dish he/she ought to consume can be improved with time if the person of concern is more serious about the matter.

We agree that this could be a hurdle for practicing the approach. In order to achieve consistent information (i.e. the mass and the maximum amount of convertible glucose) for a given dish, the cooking procedure should be repeatable. Therefore, the addition of ingredients needs to be standardized and followed by the chiefs. Some specialized training on standardized cooking can be performed to the chiefs, which is meaningful to achieve consistent information for a given dish thereby facilitating the practice of this approach in reality.

The missing word “are” has been added as suggested.

Note that for a given dish, the information printed on the menu of a certain restaurant does not necessarily have to be the same as that of another restaurant. Such a menu is only applicable within the given restaurant because a dish with the same name may be made of very different compositions in different restaurants. From the view of practicing the approach proposed in this perspective, it is not necessary to test and identify the differences between the dish with the same name prepared in different restaurants.

The approach proposed here focuses on the conservative estimate of the peak blood glucose level after the consumption of a carbohydrate dish or food product, which is closely associated with the risk of diabetes. Over 10% of the populations in China alone are diabetic. The diabetic population in China and that in India combined, makes the world’s 3rd largest country as such. Once diagnosed with diabetic condition, the life quality of the individual will significantly deteriorate. The idea shown in this paper, once adopted one way or another in practice, will help many people. This is why we focus on diabetes in this work. However, this doesn’t mean that the food labelling information regarding diabetes is more important than the other health information for other diet-related diseases such as cardiovascular disease. In fact, we already stated that other health information may also be included in a menu or package in lines 348 to 354: “Furthermore, when some other health information such as the undesirable lipids etc., may also be listed for the benefits of preventing or managing metabolic chronic diseases, e.g. dyslipidemia and fatty liver disease prone individuals. The above, though written for combating diabetic conditions in particular, similar practical reasoning may also be beneficial for people who are interested in weight control or in competitive sports”.